# End-to-End Spermatozoid Detection in Cytology WSI for Forensic Pathology Workflow

**Rutger H.J. Fick**\*                                                    RFICK@TRIBUN.HEALTH
**Mélanie Lubrano**\*                                                  MLUBRANO@TRIBUN.HEALTH
**Damien Quignon**†                    DAMIEN.QUIGNON@GENDARMERIE.INTERIEUR.GOUV.FR
**Didier Empis**†                         DIDIER.EMPIS@GENDARMERIE.INTERIEUR.GOUV.FR
**Fabrice Kabile**†                    FABRICE.KABILE@GENDARMERIE.INTERIEUR.GOUV.FR
**Saima Ben Hadj**\*                                                  SBENHADJ@TRIBUN.HEALTH

**Editors:** Accepted for publication at MIDL 2023

## Abstract

This study aimed to improve the sensitivity and throughput of spermatozoid screening for identifying rape suspects through DNA profiling, based on microscope cytology Whole Slide Imaging (WSI). To this end, we implemented a WSI-based deep-learning algorithm consisting of a detector/classification ensemble, achieving a mean 3-fold cross-validation F1 score of 0.87 [0.87-0.88] on a dataset of 188 retrospective single-center cytology WSI. Applied to slide label-only annotated test set (positive, negative, and doubtful), we show that our ensemble model is capable of screening slide label groups with excellent sensitivity to even find missed spermatozoids in negative-labeled slides. We hope our approach will be of value for routine forensic spermatozoid screening.

**Keywords:** Spermatozoid screening, Whole Slide Images (WSI), Deep learning, Forensic laboratories, Object detection

## 1. Introduction

Spermatozoid detection via cytological slide examination is a standard procedure in forensic laboratories, aiming to identify sexual assault suspects through DNA analysis of sperm retrieved from victims. The conventional microscope-based evaluation of slides is labor-intensive and susceptible to false-negative labeling, particularly when only a single spermatozoid is present. In order to enhance the screening sensitivity and efficiency, we have integrated a deep learning-driven digital method for detecting spermatozoids on Whole Slide Images (WSI) of cytology specimens.

Numerous guidelines outline the ideal methods for collecting potential sperm-containing sample (Suttipasit, 2019), as well as various techniques to obtain forensic evidence for the presence of spermatozoids (Stefanidou et al., 2010). In the context of microscope-based spermatozoid detection, the scarcity of spermatozoids and the diverse origins of sample collection (e.g., vagina, anus, hair, clothing) lead to a wide range of debris present in WSI, complicating the accurate detection of spermatozoids (see, for example, Figure 1 Given that the primary objective of screening is to identify even a single spermatozoid

---

\* Tribun Health, Paris, France
† Institut de Recherche Criminelle, Gendarmerie Nationale, France

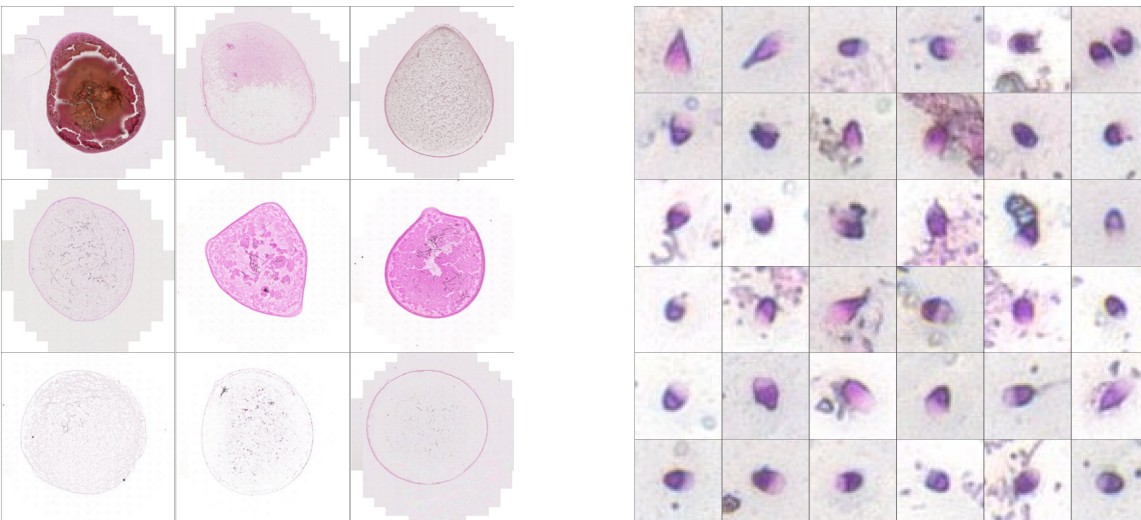

Figure 1: Left: Example spot macro-images. Right: Example sperm objects showing debris.

within the WSI, our deep learning-based approach demonstrates its ability to locate rare and previously overlooked spermatozoids, thereby proving valuable for incorporation into forensic workflows.

## 2. Materials and Methods

In this section we describe the dataset creation in Section 2.1 and training approach for our end-to-end spermatozoid detection algorithm in Section 2.2.

### 2.1. Data

Our dataset consists of 188 retrospective single-center cytology WSI – 50 for training and 138 for testing – containing HE-stained samples samples recovered from a representative source variety. We used a 2-step approach for annotating spermatozoids, first having a "weak" model trained on manual annotations detect spermatozoid candidates. Then these candidates were validated using through a 2+1 expert consensus, resulting in 6425 annotated spermatozoids and 12464 (hard) negatives. The test set is split between 45 positive, 42 doubtful, and 51 negative slide labels. Note that a doubtful classification just means that the reader suspects some objects in the WSI are spermatozoids but is not sure, and leaves the final decision of doing a DNA profiling to a secondary reader. The average spermatozoid content of negative, doubtful, and positive WSI is expected to be none, low and high, respectively. In Figure 1 we show an illustration of some representative cytology spots and annotated spermatozoids.

### 2.2. Model Training

We trained a detector/classification ensemble model for object detection, combining a Yolo-RD6 detector (Wang et al., 2022) and EfficientNet-B7 classifier (Tan and Le, 2019). The

|  | F1 Score | | |
| Training | Detector | Classifier | Ensemble |
| --- | --- | --- | --- |
| 1 | 0.80 | 0.84 | 0.87 |
| 2 | 0.81 | 0.85 | 0.88 |
| 3 | 0.79 | 0.84 | 0.87 |

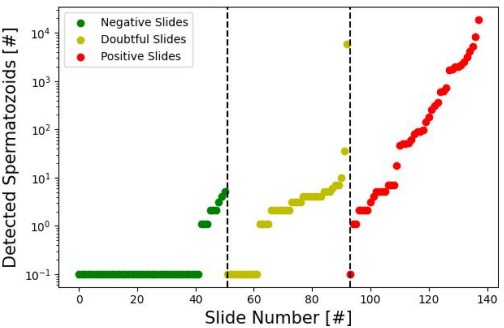

Figure 2: Left: F1 scores of three cross-validation trainings. Right: Test set evaluation, showing per slide label group how many spermatozoids were detected in logarithmic scale.

detector model was trained by randomly sampling positive and negative examples 45% of the time and random locations within the foreground 10%. To create a strong ensemble, the detector network was then used to run inference on all the training slides and all false positive detections were then used as negatives for training the classifier model (Piansaddhayanaon et al., 2023).

## 3. Results

The ensemble model demonstrates a mean 3-fold cross-validation F1 score of 0.87 [0.87-0.88]. When applied to the test set, which only contains slide labels, our model successfully identified previously undetected spermatozoids in 5 out of 51 negative slides, confirmed spermatozoid presence in 32 out of 42 doubtful slides, and verified the presence of spermatozoids in 44 out of 45 positive slides. Figure 2 illustrates these results, emphasizing that the detection of spermatozoids in negative slides typically pertains to rare, overlooked objects within an entire WSI. An expert cytologist subsequently confirmed these findings.

In terms of practical time considerations, the digital workflow substantially reduced the processing time for a typical caseload of 50 cases, decreasing it from a week to approximately 2 hours of computation time, in addition to the verification of high-scoring objects. This reduction in time highlights the efficiency of our proposed method.

## 4. Discussion and Conclusion

Our deep-learning based, end-to-end workflow for object detection shows acceptable performance for the spermatozoid screening task. Our next step is to evaluate the routine integration of our algorithm in the standard forensic pathology workflow.

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
