# OpenReview forum: "End-to-End Spermatozoid Detection in Cytology WSI for Forensic Pathology Workflow"
_MIDL.io/2023/Short_Paper_Track — MIDL 2023 Short paper track Poster_

### Official Review · Reviewer_DvGc · 2023-04-22
**Interesting application of spermatozoid Detection**

**Rating:** 6
**Confidence:** 5

**Review:**

This paper adopted the deep learning approach for spermatozoid detection in cytology WSI. Experimental results on a decent datasets demonstrated good performance. It's interesting to see the new application of deep learning, particularly in WSI, which is tedious for experts. Overall, the paper is clearly written. More extensive comparison with exsiting approaches and human experts would be suggested.

---

### Official Review · Reviewer_6sdZ · 2023-04-24
**Add experiments to compare with existing methods**

**Rating:** 6
**Confidence:** 4

**Review:**

The paper describes a study that aimed to improve the sensitivity and throughput of spermatozoid screening for identifying rape suspects through DNA profiling, using Whole Slide Imaging (WSI) and deep learning algorithms. The authors developed a WSI-based deep-learning algorithm consisting of a detector/classification ensemble that achieved a mean 3-fold cross-validation F1 score of 0.87 on a dataset of 188 retrospective single-center cytology WSI. The study showed that the ensemble model is capable of screening slide label groups with excellent sensitivity, even finding missed spermatozoids in negative-labeled slides. The authors suggest that their approach could be of value for routine forensic spermatozoid screening.

Comments:
1. Add experiments for comparison with other methods.
2. How to handle "doubtful" cases predicted by the deep learning models?